# Discrete Time Series Clustering and Delineation: A Tree-Based Approach to Linear Temporal Logic Discovery

**Brennan Cruse, Christian Muise**

Queen's University
Kingston, Ontario, Canada
{brennan.cruse, christian.muise}@queensu.ca

## Abstract

Inferring temporal logic specifications from plan traces can offer significant insight into several aspects of planning such as goal recognition, policy summarization, and system dynamic modelling. Temporal logic specifications have the power to provide significant insights for explainability because they are capable of representing relevant patterns and partitions unique to certain groups of traces. Prior work in this area has predominantly focused on the identification of specifications that satisfy all plan traces within a set, however more recently, contrastive approaches concerning the delineation of two sets have also been established. While these approaches are effective in their defined scope, they assume the existence of only one or two behavioural clusters. In this paper, we re-imagine contrastive specification learning by proposing a novel tree generation technique which allows $k$ clusters to be discovered. By embracing a Monte Carlo node-splitting approach, our algorithm seeks balance to contrastively divide any given set of plan traces into two sets with an accompanying temporal logic specification satisfying one of the sets. Recursing this procedure, we demonstrate the effectiveness of our approach to cluster and delineate plan traces, allowing temporal logic specifications to evoke insight at each level of the resulting tree.

## 1 Introduction

Time series data represents a significant opportunity for institutions and individuals to learn from the past and present to improve the future. The prevalence of unstructured data within real-world settings, however, represents an active challenge for existing analytical techniques. This impediment is especially relevant within research areas such as goal recognition, policy summarization, and system dynamic modelling, where the shared objective is to derive meaning from observed behavior. To establish meaningful insights from unstructured time-series data, partitions and patterns must be identified to effectively differentiate observations based on temporal attributes. These insights can take the form of temporal logic specifications, where specifications are used to explain a given set of observations. Currently, however, there are no existing techniques that allow multiple sets of time-series traces to be automatically clus-

tered and differentiated. To address this challenge, we propose a novel decision-tree approach to provide structure to unstructured time-series data. By leveraging linear temporal logic, our proposed method successfully clusters an unspecified quantity of plan traces using unsupervised techniques and contrastively explains how these clusters are characterized. Representing unique temporal properties of each cluster, the identified specifications contribute to the explainability of the input set of traces by describing the distinctive interaction of subgroups. Given these developments, our novel framework represents a vital first step in providing researchers a new and more powerful approach to time-series data analysis.

Prior works in the area of plan explanations have focused on effectively summarizing a single set of traces with the goal of making sense of the output of planners. This has typically been done by determining temporal specifications that are satisfied by all traces in a given set (Yang et al. 2006; Lemieux, Park, and Beschastnikh 2015). The problem with summarization-based methods like these, however, is that temporal specifications are not designed to be relevant or interesting, merely accurate. To address this, more recent research has shifted focus towards contrastive explanations, where the research task is to automatically identify specifications that differentiate two sets of traces (Kim et al. 2019). While contrastive explanations arguably offer greater insight than summarization-based approaches because they allow trade-offs within plan rationale to be understood, the conditions required for these techniques to be applicable are quite niche. Most importantly, traditional contrastive explanation research condenses the assertion of contrast to exactly two sets of traces; in practical applications, natural decomposition likely exists between more than two clusters of traces. To accommodate this, we reimagine contrastive explanations to account for multiple sets of traces. Given the significant research interest that has been demonstrated in these similar areas in recent years, this topic is demonstrated to be both important and relevant. The main contribution of this paper includes an extension of contrastive explanations to differentiate multiple sets of traces via a decision-tree method. As a further contribution, this paper's proposed methodology allows contrastive sets to be automatically identified through a novel unsupervised clustering process.

Our methodology approaches contrastive explanations as an unsupervised learning task, where the objective is to automatically identify $k$ sets of traces guided by temporal logic specifications. We begin by formulating this as a sub-problem, where the challenge becomes identifying an efficient and accurate technique to split a single set of plan traces into two balanced sets of near-equal size via temporal logic specifications. To achieve information gain when splitting a given set of plan traces, the identified temporal logic specification must entail the first subset of traces, but not the second; the more equal these subsets are in size, the greater the information gain of the split. By proposing a solution to this sub-problem and recursing on that, our decision-tree method focuses on discovering trees of LTL specifications that allow human users to understand plan traces with respect to the behaviour of acting agents.

Applying tree generation to six benchmark planning domains of three unique vocabulary sizes, we demonstrate the generality of our algorithm to adapt to diverse problem sets. Evaluating the size of the resulting LTL specifications from this assessment, we discover a tendency of robust fit, mitigating threats of underfitting/overfitting behavior. For a deeper analysis of the mechanics of our approach, we measure the relationship between node size and splitting execution time, discovering a near-linear positive correlation and justifying our sampling step. To further highlight the effectiveness of sampling, we derive probability distributions of information gain with respect to sample size, observing consistently high information gain densities for samples as small as 10% of the original set of traces. Finally, we investigate the benefit in time of additional search iterations, discovering a trend of diminishing marginal returns, which indicates a lack of necessity for excessive iterations.

## 2 Preliminaries

### 2.1 Linear Temporal Logic (LTL)

Linear temporal logic (LTL) is a modal logic which was first proposed by Pnueli in 1977 and has become widely adopted for applications such as automata-theoretical model checking (Vardi 1996; Rozier 2011; Latvala 2005), property expression in formal verification (Grosse and Drechsler 2003; Kupferman 2006), and as a specification language (Kim et al. 2019; Kasenberg and Scheutz 2017; Lemieux, Park, and Beschastnikh 2015). Due to its powerful ability to encode relationships between events in time, LTL represents an effective medium for describing system behaviour in relation to the past, present, and future. LTL's modalities referring to time enable formulae to express concepts such as possibility, necessity, and existence. The syntax of LTL consists of two fundamental operators, "next"(X) and "until"(U). The first fundamental operator, X, is designed to provide a constraint for the next period in time, where the proposition $X\phi$ is defined to be true if in the next time period $\phi$ is true. The second fundamental operator, U, is designed to connect various fluents with each other, where the proposition $\phi\ U\psi$ is defined to be true if $\phi$ remains true in every state until $\psi$ becomes true. Using these fundamental operators, additional operators can also be created.

| Template | Meaning |
|---|---|
| global | $p_i$ is true throughout the entire trace |
| eventual | $p_i$ eventually occurs (may later become false) |
| stability | $p_i$ eventually occurs and stays true forever |
| response | If $p_i$ occurs, $p_j$ eventually follows |
| until | $p_i$ has to be true until $p_j$ eventually becomes true |
| atmostonce | Only one contiguous interval exists where $p_i$ is true |
| sometime_before | If $p_i$ occurs, $p_j$ occurred in the past |

Table 1: The set of LTL templates embraced within BayesLTL (Kim et al. 2019), adopted within our approach. When multiple propositions are used in a template, the condition is asserted for all propositions using conjunction. See BayesLTL for formal LTL translations.

When evaluating the size $|\varphi|$ of an LTL formula, we embrace Gaglione et al.'s definition, which counts the number of unique subformulas contained within an expression. For example, the size of $\varphi = (p\ UXq) \vee Xq$ is 5 because the unique subformulas in $\varphi$ are $p$, $q$, $Xq$, $p\ UXq$, and $(p\ UXq) \vee Xq$ (Gaglione et al. 2021). Since we employ BayesLTL (Kim et al. 2019) as a subprocess within our algorithm, we embrace the same interpretable templates used by Kim et al. in our research. These templates were selected within BayesLTL due to their widespread use within software verification systems; these templates can be viewed in Table 1.

### 2.2 BayesLTL

Leveraging the strengths of LTL, BayesLTL (Kim et al. 2019) proposes a method to contrastively explain the differences between two sets of plan traces using LTL specifications. BayesLTL approaches specification learning as a Bayesian inference problem by building upon the fundamental Bayes theorem $P(\varphi|X) = \frac{P(\varphi)P(X|\varphi)}{\sum_{\varphi \in \Phi} P(\varphi)P(X|\varphi)}$.

The goal of BayesLTL is to then infer $\varphi^* = argmax_\Phi P(\varphi|X)$, where $P(\varphi)$ represents the prior distribution over the hypothesis space, and $P(X|\varphi)$ is the probability of observing evidence $(\pi_A, \pi_B)$, representing two unique sets of traces, given LTL specification $\varphi$. A probabilistic generative modelling approach is then used through the development and implementation of a prior function, a likelihood function, and a proposal function. BayesLTL's prior function is built to allow the system designer to incorporate their preferences. For example, the user might choose to specify a preference for "global" operators versus "until" operators. According to the system designer's parameter configuration, the prior function chooses a LTL template from a table of potential options (shown in Table 1) and decides upon the number of conjuncts and proposition instantiations for the various conjuncts. The likelihood function asserts a contrast between the two sets of traces. By assuming that individual traces within sets are independent of each other, the likelihood of observing the input sets of traces within the satisfying $(\pi_A)$ and non-satisfying $(\pi_B)$ sets can be calculated via $P(X|\varphi) = \prod_{i=1}^{\pi_A} P(\pi_i|\varphi) \prod_{j=1}^{\pi_B} P(\pi_j|\varphi)$. Satisfaction checks are then conducted over all traces from both sets for the respective

LTL specifications, which also provides robustness for outliers and noise. Finally, BayesLTL's proposal function approximates the true posterior and MAP estimates $\{\varphi^*\}$ by sampling from the true posterior distribution and applies a Markov Chain Monte Carlo (MCMC) method to optimize template and LTL selection. In coordination with each other, these functions operate effectively together to generate relevant and interesting LTL specifications and differentiate a pair of trace sets.

While BayesLTL is highly effective at identifying accurate and relevant contrastive explanations, the assertion of contrast is limited to exactly two sets of traces. In consideration of this limitation, our framework expands contrastive analysis to allow $k$ sets of traces to be evaluated. Additionally, BayesLTL requires input sets to be predefined and labelled, which presents another challenge for applications where associative groupings are unknown. To combat this limitation, our approach allows $k$ sets to be automatically identified via our novel clustering procedure. By expanding the scope of contrastive explanations and enabling automatic identification of clusters, our framework leverages the strengths of BayesLTL along with two novel contributions to enhance the explainable power of contrastive explanations.

## 3 Problem Statement and Approach

### 3.1 Tree and Node Structure

Our proposed delineation tree consists of nodes and edges that resemble a binary tree data structure. The primary attribute that defines a given node is a set of internally stored traces. Non-terminal nodes also possess an LTL formula that is catered to their respective traces and designed for delineation. By using a given node's LTL specification, traces within that node can be contrastively evaluated based on entailment. This evaluation allows two new nodes to be created, whereby traces are allocated into two subsets based on whether the formula is satisfied by a given trace. These subsets of traces are initialized as new nodes, and this novel process is recursively implemented and repeated until a tree is fully created. Once the tree generation process is complete, the delineation of nodes can be analyzed as a collective or in subsets of any size. By evaluating the shortest path between any pair of nodes, the conjugation of LTL along the tree's branches allows nodes to be accurately differentiated and contrastively explained. These LTL specifications effectively answer the question of what temporal property differentiates a given set of traces from one or more other sets of traces. See Figure 1 for a visual representation of this tree structure.

### 3.2 Node Splitting Criteria

As the primary engine of tree generation, our approach relies on a node splitting technique that is designed to efficiently discover contrastively differentiating LTL specifications. Representing the primary sub-problem of cluster identification and delineation, the challenge of this step is to automatically discover LTL specifications that maximize information gain. Formally, we define information gain of a specification $\varphi$, given node $n$, containing the set of traces $\pi$

as:

$$1 - \frac{\left| |\{\pi : \pi \models \varphi, \pi \in \pi_n\}| - |\{\pi : \pi \not\models \varphi, \pi \in \pi_n\}| \right|}{|\pi_n|}$$

(1)

Specifications that demonstrate information gain of 1 are considered perfect, while specifications with information gain of 0 are ineffectual. By maximizing information gain, we maximize the balance of the resulting decision tree, leading to an optimally efficient and concise data structure.

To discover specifications that maximize information gain, our technique embraces the explanatory power of the BayesLTL framework (Kim et al. 2019). Given the effectiveness of BayesLTL to identify LTL specifications differentiating two groups of traces, the tool can be used as an instrument to assist in the discovery of relevant splits. BayesLTL, however, requires positive and negative labels for input traces. Therefore, given a single set of unlabelled traces, the problem becomes identifying the optimal allocation of traces into positive and negative subsets ($\pi_+$ and $\pi_-$), such that an LTL specification can be discovered that maintains balance post-evaluation of entailment. The quantity of ways in which these two equally sized subsets can be created intuitively leads to a combinatorial explosion. To combat the complexity associated with this immense search space, we employ both sampling and Monte Carlo search.

We first use random sampling without replacement to mitigate the complexity of each search step. This means that instead of evaluating BayesLTL (Kim et al. 2019) on positive and negative sets of size $\frac{1}{2}|\pi_n|$, we reduce the contrastive evaluation to input sets of size $\frac{p}{2}|\pi_n|$, where $p$ represents the size of the sample proportion from the parent set. We demonstrate the representative abilities of varying sample proportions, and allow this value to be adjusted as a parameter. By reducing search step complexity through our use of sampling, search capacity is enhanced.

---

**Algorithm 1: LTL Specification Search**

---

**Input**: Parent set of traces $\pi_n$
**Parameters**: Iteration limit $max\_iter$, Information gain threshold $\tau$, Sample proportion $p$
**Output**: LTL specification $\varphi_{best}$

1:  $\varphi_{best} \leftarrow None$
2:  **for** $i = 0...max\_iter$ **do**
3:      $\pi_{sample} \leftarrow$ random sample set of size $p|\pi_n|$ from $\pi_n$
4:      $\pi_+, \pi_- \leftarrow$ random balanced split of $\pi_{sample}$
5:      **for** values of $\varphi$ resulting from BayesLTL($\pi_+, \pi_-$) **do**
6:          **if** $InfoGain(\varphi, \pi_n) \geq \tau$ **then**
7:              **return** $\varphi$
8:          **else if** $!\varphi_{best}$ **or** $InfoGain(\varphi, \pi_n) > InfoGain(\varphi_{best}, \pi_n)$ **then**
9:              $\varphi_{best} \leftarrow \varphi$
10:         **end if**
11:     **end for**
12: **end for**
13: **return** $\varphi_{best}$

---

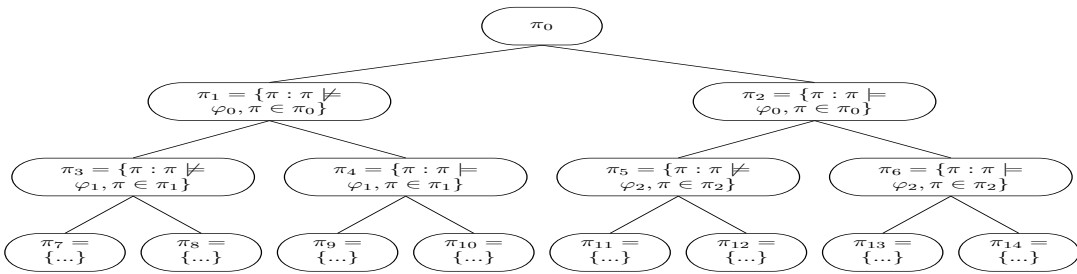

Figure 1: A visual representation of the binary tree structure used within our delineation process. Each node $\pi$ represents a collection of traces sorted by entailment for each formula $\varphi$ down the tree.

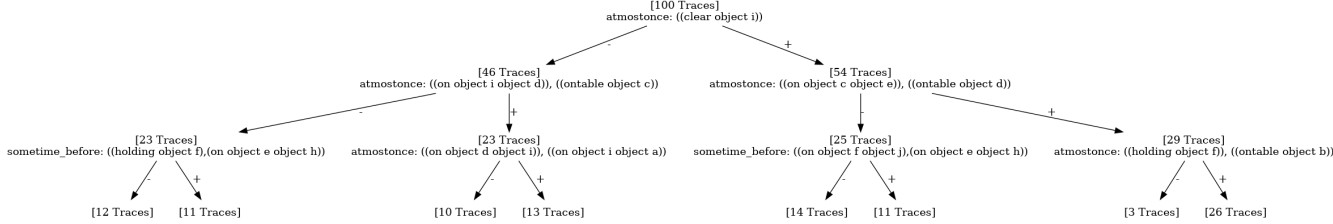

Figure 2: Example of a Generated Tree from Blocksworld Domain

We iterate search according to a Monte Carlo approach, which is procedurally shown in Algorithm 1. Our proposed approach begins by sampling a subset of traces from the parent set, as previously described. We then randomly split this subset into two equally sized new subsets, labelling one positive and the other negative. Applying BayesLTL (Kim et al. 2019) to these positive and negatively labelled sets of traces, we arrive at a list of contrastive explanations, which each individually attempts to best describe the variation between the two groups. Using this list of discovered LTL specifications, we score each formula on the information gain it offers to the parent set of traces. If a given specification is found to provide information gain that is either perfect or is beyond a parameterized threshold, the specification is accepted and the splitting process is ceased for that node. Alternatively, the specification offering the highest information gain is compared with the previous best, and the better specification is retained. This entire process is repeated until a parameter representing the maximum permitted quantity of iterations is reached, and the best LTL formula is accepted. This information gain-maximizing formula is then stored in the parent node and used as a splitting mechanism to create child nodes.

## 4 Evaluations and Results

### 4.1 Evaluation Dataset Derivation

To establish an evaluation dataset of discrete time-series data, we embrace planning as an environment for testing. Six benchmark domains were selected from International Planning Competition (Fox and Long 2003), and used in conjunction with a diversity-bounded diverse planner (Katz and Sohrabi 2020). Representing assortment in testing, these selected domains include Blocksworld, Gripper, Rovers, Satel-

lite, TPP, and ZenoTravel. Additionally, within each domain, three vocabulary sizes $|V|$ in $\{10, 15, 20\}$ were used. Given these 18 unique configurations, 100 diverse plans were generated for each configuration using a stability similarity threshold (Fox et al. 2006; Coman and Munoz-Avila 2011) of 0.25, enforcing every possible pair of two plans to have a maximum of 25% identical actions. By following this process, we arrived at 18 unique datasets to probe various aspects of our methodology in testing.

Similar to Kim et al.'s evaluation of BayesLTL (Kim et al. 2019), we also explored injection of LTL ground truth into problem/domain PDDL files via LtlFond2Fond (Camacho et al. 2017). However, this additional step was deemed unnecessary for our purposes, since preliminary results demonstrated an abundance of naturally balancing partitions existing between diverse plans generated using stability-similarity. Unlike BayesLTL where two sets of traces are contrasted, our approach contrasts multiple sets of traces, resulting in a greater quantity of available solutions. Since several accurate ground truths exist within our generated diverse plans, the expectation for any specific predetermined solution to be found is typically unreasonable.

### 4.2 General Effectiveness of Tree Generation

Since our approach merely represents an introduction to the solution of this problem, functional testing is beyond the scope of this paper. We, therefore, leave functional testing to future work and instead focus our evaluation on the inner mechanics and properties of our proposed solution. Additionally, since we are the first to expand contrastive explanations to $k$ automatically identified groups, there is no existing benchmark for us to relatively measure our success. We must therefore rely on internal metrics to assess the strength

| Domain (100 Traces) | $\|V\|$ | Average Depth (20 Trees) | | Average Info Gain (20 Trees), ($\|\pi\| \geq 10$) | |
|---|---|---|---|---|---|
| | | Med | Mean | Med | Mean |
| Blocks | 10 | 6.299 | 6.308 | 0.505 | 0.501 |
| | 15 | 6.372 | 6.404 | 0.388 | 0.380 |
| | 20 | 6.219 | 6.202 | 0.128 | 0.135 |
| Gripper | 10 | 5.789 | 5.792 | 0.860 | 0.867 |
| | 15 | 5.995 | 6.009 | 0.729 | 0.726 |
| | 20 | 6.382 | 6.381 | 0.622 | 0.618 |
| Rovers | 10 | 5.836 | 5.821 | 0.788 | 0.783 |
| | 15 | 5.942 | 5.944 | 0.873 | 0.876 |
| | 20 | 5.783 | 5.776 | 0.879 | 0.880 |
| Satellite | 10 | 5.789 | 5.788 | 0.911 | 0.921 |
| | 15 | 5.810 | 5.811 | 0.844 | 0.847 |
| | 20 | 5.783 | 5.797 | 0.879 | 0.873 |
| TPP | 10 | 5.779 | 5.784 | 0.897 | 0.898 |
| | 15 | 5.789 | 5.787 | 0.891 | 0.893 |
| | 20 | 5.799 | 5.796 | 0.888 | 0.893 |
| Zeno-Travel | 10 | 5.779 | 5.781 | 0.898 | 0.899 |
| | 15 | 5.784 | 5.784 | 0.897 | 0.895 |
| | 20 | 5.779 | 5.779 | 0.899 | 0.895 |

Table 2: Performance results of 20 test cases for each of the listed domains and vocabulary sizes $|V|$. Trees were generated using a sample proportion of 40%, an information gain threshold of 80%, and a maximum of 10 iterations, isolating all unique traces as leaves. Each row reports the median and mean of average node depth and average information gain from the 20 samples of each respective domain/vocabulary configuration. For reference, a perfect tree initialized with 100 traces and possessing maximal information gain at every node, would have an average node depth of 5.759.

| Domain (100 Traces) | $\|V\|$ | Average LTL Size (20 Trees), ($\|\pi\| \geq 10$) | | | |
|---|---|---|---|---|---|
| | | Min | Med | Max | $\sigma$ |
| Blocks | 10 | 6.905 | 7.841 | 8.857 | 0.626 |
| | 15 | 4.381 | 5.390 | 6.947 | 0.676 |
| | 20 | 3.600 | 4.667 | 5.733 | 0.562 |
| Gripper | 10 | 5.357 | 6.942 | 8.357 | 0.765 |
| | 15 | 6.933 | 8.036 | 9.063 | 0.742 |
| | 20 | 7.125 | 8.092 | 10.222 | 0.744 |
| Rovers | 10 | 5.824 | 7.129 | 10.467 | 1.081 |
| | 15 | 7.071 | 8.171 | 10.467 | 1.083 |
| | 20 | 5.400 | 8.157 | 10.800 | 1.231 |
| Satellite | 10 | 6.266 | 7.633 | 10.231 | 1.104 |
| | 15 | 4.643 | 6.829 | 8.692 | 0.712 |
| | 20 | 5.214 | 6.379 | 8.067 | 1.009 |
| TPP | 10 | 6.423 | 8.100 | 10.307 | 1.104 |
| | 15 | 4.643 | 7.031 | 11.214 | 1.601 |
| | 20 | 5.214 | 6.893 | 9.286 | 0.935 |
| Zeno-Travel | 10 | 5.857 | 9.033 | 11.400 | 1.407 |
| | 15 | 6.308 | 8.136 | 10.357 | 1.180 |
| | 20 | 4.923 | 7.833 | 9.786 | 1.085 |

Table 3: Analysis of LTL specification size $|\varphi|$ distribution within the test cases established in Table 2. LTL size is measured according to the quantity of unique subformulas, as defined by Gaglione et al.. Each row reports the minimum, median, maximum, and standard deviation of average $|\varphi|$ within the 20 sample trees of each respective domain/vocabulary configuration, where robust fit is apparent.

of our approach.

To evaluate the effectiveness and applicability of our delineation process, we analyze our approach's adaptability and scalability potential. To assess these attributes, we generated 20 trees for each domain/vocabulary size configuration in our evaluation dataset. An example of one of these trees can be viewed in Figure 2, where we display four levels of a contrastive explanation tree derived from the blocks domain with a vocabulary size of 10. Post-generation, we then judged the strength of the discovered trees according to balance and information gain metrics. Balance, which represents the efficiency of our process to organize and differentiate traces, was calculated using average node depth. Since high quality specifications minimize the number of splits required to isolate traces, smaller values of average depth are desirable, as they are indicative of higher quality specifications. When analyzing average depth of trees, it is important to note they can both only be evaluated in contrast to trees with identical quantities of root traces, as is the case within our presented experiment. Alternatively, the balance can also be evaluated as a ratio of $\log_2(n)$, where $n$ represents the total quantity of nodes in a tree, releasing the measurement from its context dependency, but the resulting value is less intuitive to comprehend. We calculated information gain according to Formula 1 for all nodes containing at least 10 traces in each tree, then averaged these values for each individual tree, presenting the median tree. The minimum node size of 10 traces was selected as a filter for our

analysis of information gain as averages should not be overshadowed and overstated by including easier splits of less relevant nodes. By generating and evaluating several trees of diverse configurations, the resilience of our algorithm to identify high-quality specifications was demonstrated.

Table 2 summarizes the characteristics of the resulting trees generated to evaluate the efficacy of our approach. From the recorded measurements, it is clear that our algorithm is capable of discovering and differentiating clusters existing within a variety of unique domains associated with diverse vocabulary sizes. In analyzing the median information gain of our evaluation trees, we observe little impediment associated with higher-complexity problems. This is also shown to translate successfully to median average depth, where measures appear similarly small across all domains and vocabulary sizes. These observations effectively demonstrate our algorithm's ability to cluster and delineate traces in an efficient manner.

### 4.3 Analysis of Discovered Specifications

Since the interestingness of a given specification is relative to unique problems, interestingness is not something that can be quantitatively evaluated or generalized. However, since we rely on the BayesLTL framework (Kim et al. 2019) as a subprocess in our LTL discovery method, we can also rely

| Domain | LTL Specification ($\varphi$) | Size ($|\varphi|$) |
|---|---|---|
| Blocks | atmostonce: ((clear object i)), ((on object a object d)), ((on object j object b)) | 19 |
| Gripper | eventual: ((carry object ball16 object right)), ((carry object ball19 object left)), ((carry object ball21 object right)), ((carry object ball4 object left)) | 9 |
| Rovers | atmostonce: ((calibrated camera camera2 rover rover1)), ((calibrated camera camera5 rover rover3)) | 13 |
| Satellite | sometime_before: ((have_image object star15 object spectrograph1), (pointing object satellite4 object phenomenon8)) | 5 |
| TPP | eventual: ((loaded goods goods5 truck truck2 level level1)) | 2 |
| Zeno-Travel | eventual: ((at object person3 object city4)), ((fuel-level object plane1 object fl1)), ((in object person8 object plane2)) | 7 |

Table 4: Examples of LTL specifications discovered within trees of our six evaluation domains, along with their respective measurements of size. See Table 1 for descriptions of templates. We measure LTL size according to the number of unique subformulas within a given expression, as defined by Gaglione et al..

on the success that BayesLTL presents in discovering relevant specifications based on the preferences of the user.

Quantitatively though, it has been shown that our algorithm is capable of identifying high-information gain LTL specifications to establish minimal-depth trees; however, a more extensive analysis of those specifications is necessary to rule out unintended behaviour. For instance, it would likely be problematic if only large specifications were identified because this would be indicative of overfitting, mitigating the usefulness of discovered formulae. To investigate the quality tendency of discovered specifications, we evaluate the size of the specifications within our sample trees; we measure the size according to the number of unique subformulas within a given expression, as defined by Gaglione et al..

In Table 3 we present the resulting size measurements of specifications within our 20 evaluation trees for each domain/vocabulary configuration. These values positively demonstrate our algorithm's ability to identify low-complexity formulas corresponding with a robust fit for each configuration within our test data. To provide a clear understanding of the LTL specifications our process discovers, we present example formulas in Table 4 along with their respective sizes. Another insightful finding in these measurements is the tendency of LTL size to negatively correlate with vocabulary size in most domains. This trend is likely due to the idea that higher complexity domains have greater quantities of natural ground truth to be found, which eases discovery.

### 4.4 Node Size Versus Splitting Time

Since our Monte Carlo node-splitting technique with randomized balanced subsets of traces represents the primary sub-process behind our delineation method, the execution time performance of this step, with respect to the quantity of analyzed traces, was of interest. We expected the execution time required to split a single node of traces to increase with the quantity of input traces; however, the rate of growth defining this relationship was unknown. To approximate this

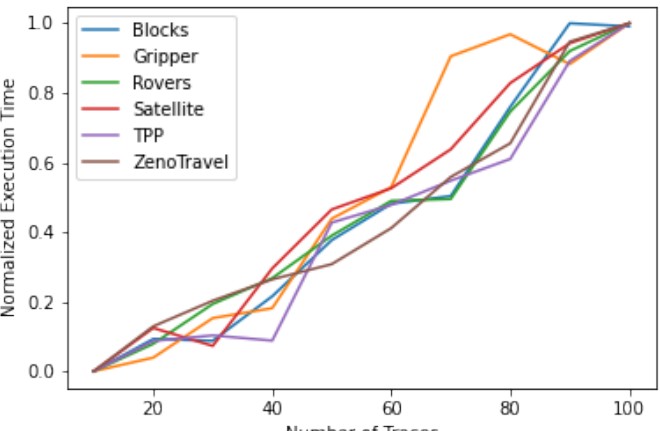

Figure 3: The average execution time of splitting a node $\pi$ of randomly sampled traces with respect to the quantity of traces $|\varphi|$ over 10 splits for each value of $|\varphi|$ in $\{10n : n \in Z^+, n < 11\}$. Since scales of execution time differ relative to each domain, normalization is conducted via $\frac{x_i - min(x)}{max(x) - min(x)}$ for plotting purposes.

rate of growth, we conducted 10 splits for each size $|\varphi|$ in $\{10n : n \in Z^+, n < 11\}$ of randomly sampled subsets of traces, measuring the execution time of each split. By plotting these data points for each domain, the relationship between node size and splitting time can be visualized.

In Figure 3, we observe a near-linear relationship between node size and splitting time within all six of the explored domains. This relationship emphasizes the importance of our sampling step since the time complexity of a single split appears to grow continuously large.

### 4.5 Probability Distribution of Information Gain with Respect to Sample Size

Given a randomly sampled subset of traces of a larger set, it is of interest to derive the probability distribution of in-

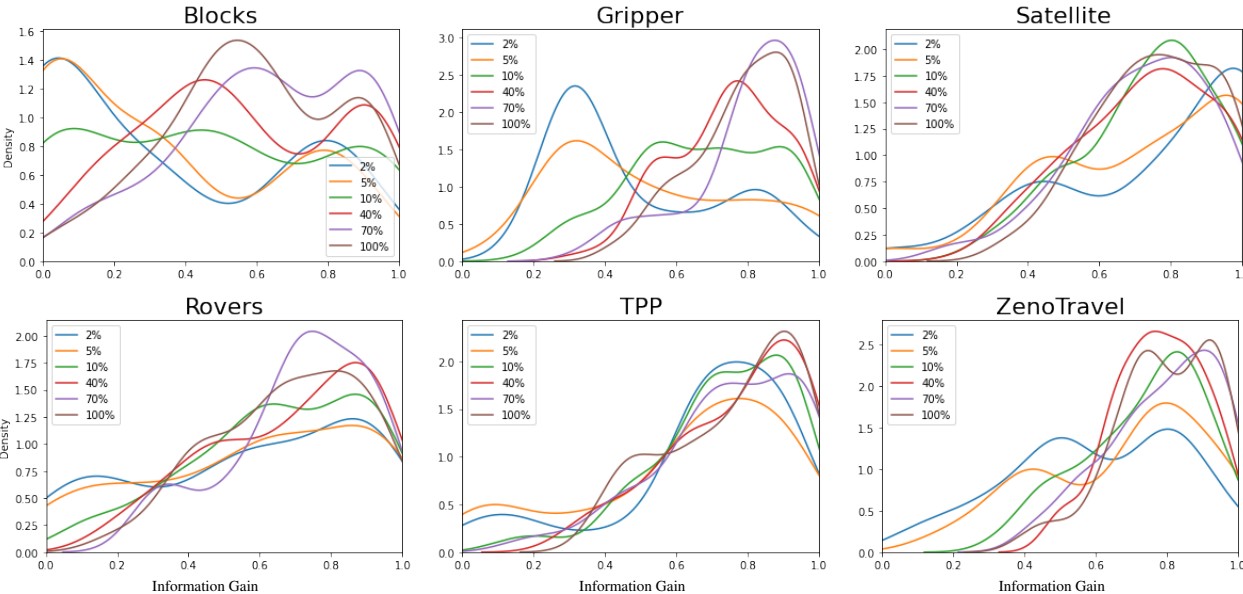

Figure 4: Kernel density estimation (KDE) plots of information gain with respect to sample proportion used when approximating split. Each subplot shows the KDEs of 100 single-split iterations for each sample proportion in $\{0.02, 0.05, 0.1, 0.4, 0.7, 1\}$.

formation gain. Intuitively, larger sample sizes should have stronger representative abilities; however, since runtime is dependent upon the size of the input set, accepting lesser probabilities may be of better utility. We estimated this distribution by running 100 splits of sample proportions in $\{0.02, 0.05, 0.1, 0.4, 0.7, 1\}$ for each domain and analyzing the resulting information gain.

As seen in Figure 4, sampling was shown to be highly effective and proved capable of representing the population distributions. Within all six of the evaluated domains and across each of the tested sample proportions, significant density was observed in the upper range of the information gain spectrum. As would be expected, higher sample proportions tended to provide greater representative ability; however, the size of this effect was interestingly small-scale. Within some domains, such as Satellite and TPP, we observe sampling effectiveness with sampling sizes as small as 2%; within all domains, however, we observe effectiveness with samples as small as 10%. This tiny performance cost of using small sample sizes represents a strong opportunity for runtime savings within complex domains.

### 4.6 Specification Exploration Time Utility

When searching for an LTL specification to split a given set of traces, the number of exploration iterations permitted will impact the information gain of the resulting split. Since iterations represent opportunities for better specifications to be found, the number of iterations should negatively correlate with the resulting information gain. The rate at which information gain is improved per iteration is of interest because iterations come at a cost of execution time. To investigate this tradeoff, we conducted 20 splits for each iteration limit value in $\{5i : i \in Z^+, i < 11\}$ for each evaluation domain, recording execution times and discovered LTL. By analyz-

ing the average information gain and average execution time of these splits with respect to the iteration limit used for each domain, the curve of this relationship can be approximated.

When increasing specification search iterations, we observe performance improvements through measurements of information gain across all six evaluation domains. However, with marginal increases, the size of observed information gain improvement tends to decrease. This trend of diminishing marginal benefit of search iterations persists across all experimented values of permitted iterations, while search time appears to increase near-linearly. This means that although the cost of search time increases at a constant rate, the marginal value received by incurring this cost decreases with higher values of the parameter. This curve of diminishing marginal value can be seen in Figure 5. This trend can likely be attributed to the idea that specifications satisfying the information gain threshold do not always exist, or, for a variety of reasons, may be less conducive to being found. Since our algorithm's precondition for early stopping is achieving a minimum score of information gain, if this milestone is impossible to reach, all iterations will still be conducted, even if a globally optimal specification has already been found.

### 4.7 Future Work

With complexity mitigation in mind, extensions focused on improving algorithmic efficiency represent the greatest potential for future work. One way this could be achieved is through an improved splitting policy, such as allowing marginal information gain to inform the number of iterations. Another avenue, which we initially investigated, is a process whereby splitting criteria could be intelligently identified via the analysis of previous splitting attempts. In this revised procedure, when a discovered specification fails to

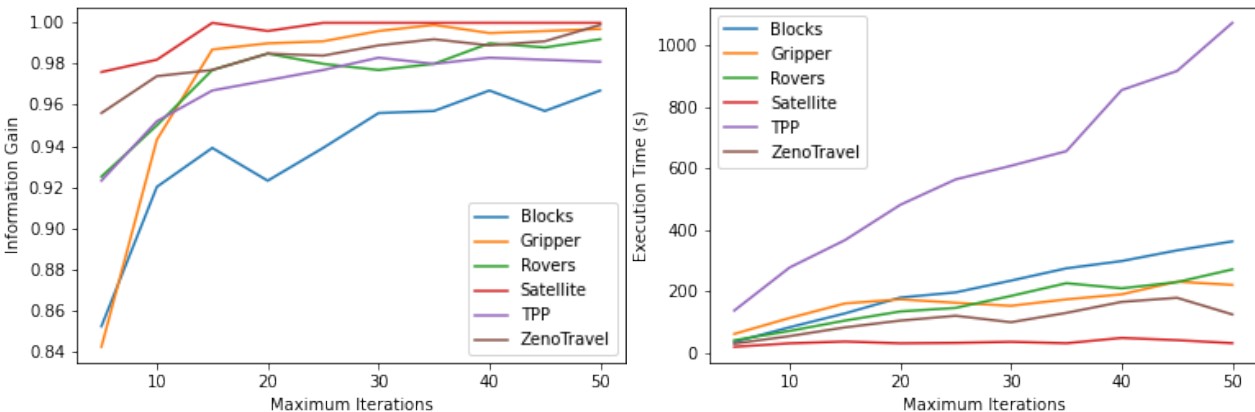

Figure 5: The average information gain and average execution time with respect to permitted maximum quantity of iterations over 20 sample splits for each test value and evaluation domain. Splits were conducted using a sample proportion of 10% with an information gain threshold of 100% to ensure maximal performance potential.

meet the information gain threshold, we do not resample $\pi_+$ and $\pi_-$ from $\pi_{parent}$. Instead, we reestablish $\pi_+$ and $\pi_-$ as the subsets of traces that satisfy and dissatisfy $\varphi$, respectfully. We then rebalance $\pi_+$ and $\pi_-$ by randomly drawing traces from the majority set and inserting them into the minority set. The hope of this revised strategy was that the sizes of these subsets would converge over iterations, leading to the desired information gain specified by the threshold.

However, when testing this revised process to split nodes of traces, we observed identical formulas being discovered repeatedly until the maximum iterations were reached. We observed this same pattern across all six of our evaluation domains, indicating a lack of success for this specific procedure. With no variety in the formulas being discovered, our revised algorithm failed to learn, meaning the resulting information gain was equal to that of the first identified formula. While this observed failure to learn rejects the proposed method as a viable alternative, it is possible that the approach could be adapted and improved to make learning feasible. From our experiment, we can deduce that in order for an intelligent-splitting algorithm to be successful, it is likely that greater freedom from the initial random allocation of traces is necessary. This could perhaps take the form of randomized resets or other statistical methods to introduce variance; we identify this side of our research to be the area with the greatest potential for future work.

## 5  Related Work

As an alternative to traditional planning research, which seeks to investigate the process of identifying optimal plans from problems, the area of plan explanation seeks to identify and describe characteristics of problems from plans. Through the analysis of observed behaviour, the goal of plan explanation is to use abductive reasoning to infer reasoning behind observed actions to better understand why certain events occur. To rationalize observed actions, plan explanation research focuses on automatically learning temporal properties that allow system behaviour to be modelled, understood, and predicted.

Plan explanations have been relevant in goal recognition settings, where they have been used as a means to infer latent goal states from incomplete observations (Ramírez and Geffner 2010; Sohrabi, Riabov, and Udrea 2016), and explicable planning research has embraced plan explanations to synthesize plans that are self-explanatory for a human's mental model (Zhang et al. 2017). Research focused on system diagnosis has also incorporated plan explanations to explain failures (Göbelbecker et al. 2010; McIlraith 1999). While each of these works focuses on identifying accurate specifications to explain plans, they focus on describing a single set of traces. Our approach, on the other hand, is designed to contrast multiple sets of traces and characterize behavioural differences.

Prior works on contrastive explanations have elevated the plan explanation problem to describe temporal differences between two sets of plan traces. Approaches to contrastive explanations have adopted SAT-based methods (Neider and Gavran 2018; Camacho and McIlraith 2021; Gaglione et al. 2021), in addition to Bayesian inference, as in the BayesLTL framework (Kim et al. 2019). Our approach embraces the Bayesian inference strategy, using BayesLTL as a subprocess. However, instead of limiting the assertion of contrast to two sets of plan traces, we evaluate contrast amongst $k$-sets. Our approach also clusters input traces to establish suitable contrastive sets automatically, as opposed to prior works, which require contrastive sets to be predefined by the user.

Leveraging decision tree learning algorithms to infer temporal logic formulas is an area that has also been previously explored (Bombara et al. 2016; Brunello, Sciavicco, and Stan 2019; Gaglione et al. 2021). As the most relevant approach in this space, Gaglione et al. use decision tree learning in their Algorithm 2 to discover contrastive LTL specifications between two sets of plan traces. Similarly, our approach adopts a decision tree learning method for contrastive explanations; however, our tree's structure is also designed to hold sets of traces in nodes, rather than only representing formulas. We use decision tree learning for cluster discovery, where we construct formulas simultaneously.

# 6 Concluding Remarks

The challenge of inferring temporal logic specifications to cluster and delineate plan traces is relevant to a wide array of planning applications. In this work, we reimagined the trace clustering and delineation challenge by proposing a novel tree generation technique which allows $k$ clusters to be automatically discovered and described. By embracing a Monte Carlo node-splitting approach, our algorithm seeks balance to contrastively divide any given set of plan traces into two sets with an accompanying temporal logic specification satisfying one of the sets. Recursing this procedure, we demonstrate the effectiveness of our approach to cluster and delineate plan traces from our evaluation dataset of benchmark domains, allowing temporal logic specifications to evoke insight at each level of the resulting decision tree.

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
