# OpenReview forum: "Discrete Time Series Clustering and Delineation: A Tree-Based Approach to Linear Temporal Logic Discovery"
_icaps-conference.org/ICAPS/2022/Workshop/XAIP — XAIP 2022_

### Official Review · Reviewer_CzvD · 2022-04-28
**Interesting Problem & Good Writing. The work feels incremental on the BayesLTL framework.**

**Rating:** 6
**Confidence:** 4

**Review:**

The paper, in summary, proposes a decision-tree based method to cluster a set of plan traces in an unsupervised fashion and yield an LTL formula for each cluster (as a consequence of tree traversal on the generated Decision Tree).

In the area of explanations and plan summarizations, having a method that generates LTL formulas that distinguish one plan from another would be beneficial, especially if the LTL formulas are concise. **The paper did a good job motivating the need for an LTL-based contrastive specification & I agree that a solution to this problem will help the AI & XAIP community**. At the same time, many factors would contribute to judging whether the generated LTL is a good summarization of a given trace, but the paper seems to be focussing on coming up with an LTL formula (with system-provided vocabulary and manually set thresholds for LTL complexity).

**Proposed Approach :**

Given a set of plan traces, the paper uses a Decision Tree approach to cluster them where a node in the tree would contain a subset of plan traces. A decision tree would require the following components,

1. Node Structure: which is as defined above
2. Split Operators: The authors utilize BayesLTL formulation & present a method to efficiently find an LTL formula to split a node into two child nodes.
3. Getting several candidate child-nodes: At the current node, the authors propose a Monte Carlo sampling to select a subset of plans and then randomly assign positive and negative labels to each plan. This is fed to the BayesLTL formulation that yields the split operator in the form of an LTL formula.
4. Selection of Best operator: They propose an information gain formula, a ratio of the count of plans traces in set1 - count in set2 by the total count of plans traces in the parent set.

They also perform some interesting evaluations to justify the effectiveness of their method :

1. On 6 IPC domains (& generate 100 diverse plans for each for evaluation)
2. Measure the avg information-gain & average depth [Average depth was an interesting measure].
3. Empirically find that the node size is linearly correlated to splitting time.
4. Marginal increase in information gain over multiple iterations. [This was again interesting & I had hoped that this measure could be used during the tree generation process]

**Pros of the work :**

1. I liked the writing of the paper, it was lucid and to the point.
2. I was impressed with the fact that random sampling could yield interesting results, especially the fact that 10% of total traces could still generate useful LTLs.
3. I liked the discussion on the size of the produced LTLs. Other measures of determining
4. It was interesting to see a future-direction algorithm, and I agree with the authors that a more intelligent splitting criteria would be a good contribution.
5. The main novel technical contribution of the work, I feel, was the Monte Carlo sampling strategy that first produces a subset of plans and then assigns positive and negative labels to the plan traces. The fact that a randomly sampled plan and further, a random assignment of labels would yield comprehensible LTL formulas is very intriguing.

**Cons of the work :**

Although I have not read many papers in the area of policy summarization, specifically that yield LTLs, it seems that the proposed approach is in its preliminary stages.

To start with, the authors claim that “... temporal specifications are not designed to be relevant of interesting ...”, I am not sure what they mean by the terms relevant or interesting or how the LTLs produced by their strategy live up to these metrics. [Other than the fact that the work is coming up with contrastive specifications.]

The authors never mentioned that their approach is, in fact a decision tree (except in the Related works section) which was surprising. Additionally, the most interesting component of a decision tree is the splitting operator & the splitting criterion - (splitting operator) which in the proposed work was taken from an earlier work called BayesLTL. In its current form, I find that the work shows the effectiveness of utilizing BayesLTL combined with random sampling for unsupervised contrastive specifications and does not possess enough novelty.

The splitting criterion is the information gain formula does not depend upon other metrics like marginal information gain (in previous iterations) or other aspects of a plan. I agree with the authors [in the future work section] that the splitting approach would benefit if adapted over iterations.

I also found that the lack of any benchmark caused me to wonder what the metrics in Table 2 of Results mean. For example, it is hard to interpret that the proposed method gives a 6.309 mean average depth in Blocksworld in an absolute sense.

Additionally, because BayesLTL is such an integral part of their proposed work, a section that gives an overview of BayesLTL would have been very helpful but was missing.

Comment :

Finally, I think that the problem the authors are trying to solve is quite interesting but the work in its current form feels incremental and more coupled to the BayesLTL framework than the authors admitted in the manuscript.

---

> ### Author Response · Authors · 2022-05-06
> **Response**
>
> Thank you for your in-depth feedback and analysis. When discussing summarization-based methods for plan explanations, when we mention, “temporal specifications are not designed to be relevant of interesting,” we are simply referring to the idea that summarization-based methods often fail to establish context. This is where contrastive explanations should be preferred, since they can explain traces in relation to other relevant traces. We will update the wording of this for better clarity. Our strategy lives up to these metrics as well because we establish contrastive explanations between multiple sets of traces, deriving an even stronger form of context.
>
> We believe that our approach is novel due to our two key contributions: 1) BayesLTL identifies contrastive explanations between only two sets of traces; we expand this analysis to allow k sets of traces to be contrasted. 2) BayesLTL requires these two sets of be predefined by the user with positive and negative labels; our approach automatically identifies these groups. Although we rely heavily on BayesLTL, we believe that these two extensions are significant enough to be considered novel, however, this is something that we will do a better job of clarifying within our revised version of the paper.
>
>  Interesting comment regarding marginal information gain as splitting criterion. For instance, if information gain is consistently not improving over additional iterations, this could be used to reduce iterations. Additionally, if information gain is continuing to improve up to the iterations parameter, we could use this to allow more iterations to achieve even stronger results. We will include this in future work.
>
> The lack of benchmark certainly is a challenge, however, since we are the first to tackle this problem, there are no existing benchmarks for us to discuss. We agree that this makes it difficult to interpret some of our metrics in an absolute sense, and were wondering if you have any suggestions on how we could make this easier to understand?
>
> Finally, a section providing an overview of BayesLTL is an excellent suggestion, and we will certainly incorporate this in our revised version.

---

### Official Review · Reviewer_kFgN · 2022-04-30
**Relevance to XAIP needs clarification**

**Rating:** 5
**Confidence:** 3

**Review:**

The manuscript describes a decision-tree learning approach to clustering time series / plans. The splits at each node in the tree are done based on LTL formulae that separate the set of plans at that node into two subsets while optimizing an information-gain measure.

The manuscript is a very interesting read, but I am not convinced of its fit for XAIP. It would have been helpful to include some explicit discussion about which questions the LTL formulae are supposed to be answers to (i.e., explanations). There is an attempt in the Related Work section to establish a connection to contrastive explanations. However, a mere property that distinguishes two plans is not automatically an explanation for "why certain events occur". For instance, consider two plans both fulfilling goal p. The first plan results in p, q, the second plan in p, not-q. Thus, eventually(q) distinguishes both plans. What does this formula explain beyond that the two plans are different; what why-question is this formula an explanation to?

Pro:
- Interesting topic
- Well written
- Experimental evaluation of the algorithms included

Con:
- Connection to XAI should be stronger, relevance to workshop theme too unclear

---

> ### Author Response · Authors · 2022-05-06
> **Response**
>
> Thank you for your valuable feedback and suggestions. We understand your concern regarding relevance to XAIP and will certainly address this in our revised version of the paper. We believe that the relevance to XAIP lies in the ability of our approach to differentiate multiple sets of traces, and to do so without predefined groupings. From an unordered bag of traces, we believe that the identification of order via temporal properties represents a form of explainability. The explainable power of our approach is similar to that of existing contrastive explanation research; however, our strategy arguably provides stronger insights, due to differentiation between more than two contrastive sets. As per your example, the specification, eventually(q), would be an explanation as to what differentiates these two plans; given an abundance of traces with multiple fluents, this may be very difficult to infer without the use of a contrastive explanation framework. The meaning of q, however, in the context of the problem would be an important aspect to consider. Additionally, something that perhaps we should incorporate in the paper, is the ability that exists within BayesLTL to incorporate a user’s preference for certain templates over others. This ability allows the user to select templates that are most relevant to the problem at hand. A section on BayesLTL also came up in the other two reviews, so a better discussion on the explainability of BayesLTL is something we will include here, which should also provide better support for our approach’s relevance.

---

### Meta-Review · Program_Chairs · 2022-04-30

**Recommendation:** Accept
**Confidence:** 4

**Metareview:**

The paper presents a decision-tree approach for clustering plan traces and delineate them using linear temporal logic.

Both reviewers agree that the paper is well-written, it tackles a very interesting problem, and the technical methodology is sound. However, the reviewers also indicate two major problems with the paper: 1) the connection to XAIP is unclear; and 2) the work appears to be preliminary and incremental. Generally, the paper’s topic concerns plan summarization, which falls within the umbrella of XAIP, and a preliminary/incremental work is within the scopes of a workshop program. As such, in light of keeping the workshop a fruitful venue for discussion and dissemination of information, I am recommending accepting the paper.

Nonetheless, we urge to authors to reflect on the reviewer’s comments and address them in their revised version of the paper. Specifically, we hope the authors make a clear connection of their method to XAIP. Additionally, as the proposed approach builds heavily on the BayesLTL framework, the authors should include an overview of that framework and also explain the novelty of their approach compared to BayesLTL.

---

> ### Author Response · Authors · 2022-05-06
> **Response**
>
> Thank you very much for your feedback and recommendation for acceptance. We will reflect on the reviewers’ comments and improve the paper to better clarify its relevance to XAIP. We will also incorporate a stronger overview of BayesLTL, and a better discussion of our contributions to ensure they are better communicated. Thank you again for your helpful input.

---

### Decision · Program_Chairs · 2022-04-30

Accept